# Children’s Usage of Inclusive Playgrounds: A Naturalistic Observation Study of Play

**DOI:** 10.3390/ijerph192013648

**Published:** 2022-10-21

**Authors:** Maeghan E. James, Emma Jianopoulos, Timothy Ross, Ron Buliung, Kelly P. Arbour-Nicitopoulos

**Affiliations:** 1Faculty of Kinesiology and Physical Education, Mental Health and Physical Activity Research Centre, University of Toronto, Toronto, ON M5S 1A4, Canada; 2Bloorview Research Institute, Holland Bloorview Kids Rehabilitation Hospital, Toronto, ON M4G 1R8, Canada; 3Department of Geography and Planning, University of Toronto, Toronto, ON M5S 1A4, Canada; 4Rehabilitation Sciences Institute, University of Toronto, Toronto, ON M5S 1A4, Canada; 5Department of Geography, Geomatics and Environment, University of Toronto Mississauga, Mississauga, ON L5L 1C6, Canada

**Keywords:** play, inclusion, inclusive playground, social development, play behaviours

## Abstract

Inclusive playgrounds that are designed to be physically accessible and welcoming to children with disabilities may provide equal and equitable access to play for all children. Using a naturalistic observational design, this study examines children’s use of a playground designed to be accessible and inclusive for all ages and abilities. A modified version of the System for Observing Play and Recreation in Communities was used to collect child data on observed gender, age, play behaviour types, social interactions, and activity levels. A relatively equal number of female (52%) and male (48%) observations was made, and the majority (96%) of children observed appeared to be under 12 years of age. Most children (71%) were observed to be engaging in active play. Functional play (e.g., climbing, swinging, running) was the predominant play behaviour observed on the playground (88%), and the majority of social interactions were with peers (48%) or an adult (26%). These findings provide information on how children use a playground designed to be inclusive for children of all ages and abilities. This information can be used to help inform the design of inclusive play spaces as well as types of programming that may occur within such settings.

## 1. Introduction

Engaging in play is understood to be an important contributor to child health and development [1] and a fundamental right for all children [2]. Despite the lack of consensus on how play is defined, it is understood to involve activities that are freely chosen, child-led, opportunistic, fun, and can be experienced independently as well as with others [3,4]. Play provides opportunities for children to develop social, cognitive, and physical skills, and these are further enhanced with opportunities for interactive play [5,6,7,8,9,10]. While play can occur both indoors and outdoors, engaging in outdoor play has additional benefits to child development. Outdoor play has been shown to increase learning opportunities [11,12], enhance cooperation and decrease conflict between peers [13], and may increase physical activity and reduce sedentary behaviours [14]. Recognizing the importance of outdoor play, organizations such as Outdoor Play Canada [15] have released positional statements that recommend children have equal and equitable access to active play opportunities outdoors and in nature. 

Most playgrounds serve as community spaces that offer children opportunities to engage in outdoor, unstructured play with limited supervision from adults. Opportunities to engage in unstructured play have been shown to be positively associated with improvements in social skills and reductions in stress and anxiety [5,6,16,17]. In some instances, playgrounds also provide increased opportunities for risky play which is associated with further physical and social benefits [18,19]. Moreover, playgrounds offer a space for families to gather together which allows for children to engage in social play and interact with peers, further contributing to social and cognitive development [7,8,9,10]. In addition to social benefits, playgrounds can provide children with physical benefits. Research shows that children who have access to playgrounds engage in more physical activity [20,21] and display an increase in gross motor skills [22]. Both physical activity and motor skills have been linked to improved fitness, reduced risk for disease, and improved mental health in children [23,24]. 

Research on traditional playgrounds suggest that the selection and layout of play equipment may influence how children play on the playground. For example, boys have been shown to prefer playgrounds that offer sport courts and traditional ball games whereas girls prefer activities that engage their imagination, involve creating things, and activities like climbing and sliding [25]. With regard to activity level, Dowda et al. [21] found that children on playgrounds with unfixed play equipment (e.g., tricycles, balls) were significantly more active than children on playgrounds with only fixed play equipment (e.g., monkey bars, slides). Moreover, studies have shown that children tend to engage in a larger variety of play behaviours (e.g., gross motor skill activities, dramatic play) on playgrounds that include natural elements (e.g., tree stumps, boulders, logs) compared to playgrounds with traditional features (e.g., fixed play structures consisting of a combination of slides, climbing structures, and swings) [26,27,28]. Importantly, research on traditional playgrounds highlight that the design and layout of a playground also influences *who* can play. For example, traditional playgrounds can present physical barriers to play, often resulting in children with disabilities experiencing marginalization as part of their playground experiences. In some cases, children with disabilities may be excluded entirely as a result of playground design (e.g., when a child who uses a wheelchair comes across a playground with a sand or pea gravel surface) [29,30,31]. Recognizing the importance of offering equal and equitable play opportunities for all children, regardless of age, gender or ability level, communities continue to advocate for the construction of inclusive playgrounds. 

Inclusive playground design goes beyond accessible design (i.e., enabling users to access and travel through their play spaces freely [32]) by ensuring that playgrounds offer physical accessibility and rich play opportunities that meet the physical, social and cognitive needs of all children [30,31]. An inclusive playground has been defined as a space that allows children of all ages and genders, both with and without disabilities, to access the playground and play together, and allows families to engage in play with their children [32,33,34]. Inclusive playgrounds should support and welcome diverse bodies and play opportunities. Providing a variety of play equipment that facilitates challenging play and the opportunity to engage in various play behaviours may be one way of facilitating inclusion on playgrounds [32,35]. For example, in addition to the more common play structures found on traditional playgrounds (e.g., slides, swings, monkey bars), inclusive playgrounds may also include elements of sensory play (e.g., music elements, visual and tactile stimuli) to engage children with high sensory needs and recognizable objects to encourage imaginative play [32]. Inclusive play opportunities should allow for children to play with adults and peers but should also support solitary play for a child who, for example, is experiencing over-stimulation or is seeking a place to retreat [32,34]. By including a variety of play elements that cater to different age groups, play preferences and abilities, inclusive playgrounds can provide a welcoming space that facilitates equal and equitable play opportunities for all children.

To date, only one study to our knowledge has examined children’s experiences on inclusive playgrounds located in Switzerland [36]. This study indicated that while children with and without disabilities perceived the playground as ‘fun’ and ‘cool’, invisible social barriers still existed and inclusion on playgrounds remains complex [36]. In Canada, the number of inclusive playgrounds built in communities has increased, but there remains a notable gap in the literature regarding who is utilizing these play spaces and if they are adequately supporting inclusive play (e.g., if they provide various play opportunities, promote and enable play for children of all abilities, age groups and genders). Without a rich understanding of how inclusive playgrounds are used, it is difficult for municipalities to obtain funding for building new inclusive playgrounds or modifying existing traditional playgrounds to enhance inclusion. Understanding how inclusive playgrounds are being used by children can provide valuable insight into how these playgrounds support (or, are not supporting) different types of play behaviours and social interactions to facilitate meaningful play experiences for all children. Addressing this research gap is important for understanding how inclusive playgrounds are utilized within communities and for determining best practices for playground designers and recreation staff to ensure playgrounds are both accessible *and* inclusive play spaces for all children. In doing so, playgrounds can appeal to the needs, capabilities, and interests of all children, and inevitably provide all children, regardless of age, gender and ability, the opportunity to play.

The purpose of this study was to examine how a newly built playground intentionally designed to be accessible and inclusive of children of all ages, genders and abilities is being used by the community and by whom. We sought to describe the usage of this inclusive playground in terms of observed gender, age group and use of a wheelchair, mobility aid and/or visual aid. The types of play behaviour and social interactions, as well as activity levels of children using the playground were also examined to further understand children’s usage of inclusive playgrounds.

## 2. Methods

### 2.1. Study Design and Setting

#### 2.1.1. Study Design

This study was a naturalistic observational study whereby observations were made of children’s naturally occurring play on an accessible and inclusive playground. All children using the playground during the observation periods were included in this study.

#### 2.1.2. Setting

Observations took place at a playground specifically designed to be accessible and inclusive for children of varying abilities and developmental stages. This playground was approximately 15,000 square feet and opened to the public in Fall 2018. The playground was situated within a larger recreational park that consists of walking trails, open fields, a community centre, and designated picnic areas. Paved pathways provided access to the park and playground, and on-site parking was available.

The playground was designed and built around four thematic sections: music/sensory, early childhood, freestanding equipment, and the main structure (see Figure 1, Figure 2, Figure 3 and Figure 4, respectively). The music/sensory section allowed families and children to explore music and sensory play. It consisted of drums, chimes, and a xylophone, as well as a freestanding structure with different wheels, mirrors, and gears that serve as a tactile and sensory play centre. The early childhood section was designed for preschool-aged children and included a smaller scale play structure with a slide and climbers. The section also included a “cozy dome”—a dome-like structure that could be used for climbing, playing inside, or, for children with sensory needs, it could serve as a welcoming space with reduced sensory stimulation. The freestanding equipment section was intentionally designed as a space for children to play in pairs or small groups. It consisted of the playground’s freestanding structures, including three types of swings (i.e., a friendship swing, a belt seat swing, and a moulded bucket seat swing with harness), a large rotating globe-like structure that supports climbing, and “see-saw” structures. The main structure was the largest section. It included several different types of monkey bars, ladders, and slides, all of which are accessible via double-wide ramps and connected by a double-wide elevated pathway loop. The double-wide ramps and pathway loop allowed children using mobility devices to access and comfortably navigate the elevated play structure. In addition, the main structure included several interactive game panels (e.g., seek and find games, X’s and O’s) as well as sensory panels. Further details on the playground’s components can be provided by the corresponding author.

### 2.2. Observation Tool and Protocols

#### 2.2.1. Observation Tool

Observations were carried out using a modified version of the System for Observing Play and Recreation in Children (SOPARC) tool that was designed for use on an iPad. SOPARC has been deemed both a reliable and valid measure of children’s play [37]. SOPARC uses momentary time sampling techniques through periodic scans of pre-determined target areas to gain information regarding perceived gender, race/ethnicity, age, and activity levels [38]. Perceived gender (male, female), age (i.e., 0–5 years, 6–12 years, 13 years and older), and activity level (i.e., non-active, active, very active) were ascribed to children using the playground in the current study using the original SOPARC tool. The original race/ethnicity variable of SOPARC was not used due to the research team’s concerns with assigning a race/ethnicity to a child based solely on observation, without being able to ask them how they identify [39,40]. Given our focus on inclusive playgrounds, a variable was added to the observational tool that coded for a child using a wheelchair, mobility aid (e.g., a walker) or visual aid (e.g., a white cane). If the child being observed did not appear to use any of these devices, they were coded as ‘none’. While we recognize the value and importance of understanding how playgrounds are being used by children who may have an invisible disability (i.e., an impairment or medical condition that is not readily observable, such as autism spectrum disorder), this was a naturalistic study whereby we did not ask children or parents for any type of demographic information. As such, we were limited to only code for features that could be observed.

Social interaction type and play behaviour type were also incorporated as variables into this study’s modified version of the SOPARC tool. Social interactions were coded based on previous research examining play behaviours and social interactions on playgrounds and included peer play, play with an adult, solitary play, and parallel play [41]. Drawing on definitions used by Maxwell et al. [41], we divided play behaviour into six types: (1) constructive, (2) dramatic/fantasy, (3) functional, (4) games with rules, (5) non-play, and (6) waiting. Table 1 provides a summary of the observation variables and coding definitions for observed assistive device/aid, activity level, social interactions, and play behaviours that were incorporated into the modified SOPARC tool.

#### 2.2.2. Systematic Observation Protocol

Two researchers were trained to systematically apply the modified SOPARC tool across nine playground zones that the study team identified a priori. These zones were created to assist with observations and were sorted into their corresponding playground section during data analysis. SOPARC guidelines recommend dividing larger zones into 3–4 smaller subzones to make observations easier [37]. Therefore, each of the nine zones were split into 3–4 subzones based on the types and groupings of equipment pieces. Before systematic observations commenced, the two researchers completed two reliability sessions. This involved simultaneously conducting observations across the pre-determined zones and subzones. After each session, the researchers met to discuss any discrepancies to improve consistency of coding. Intra-class coefficients (ICC) were used to assess reliability and interpreted as follows: poor reliability (ICC < 0.50), moderate reliability (ICC = 0.50 < 0.75), good reliability (ICC = 0.75 to 0.90), and excellent reliability (ICC > 0.90) [43]. An inter-rater reliability of 0.75 (i.e., good reliability) or above on each item of the tool was achieved prior to commencing study observations. ICC values ranged from 0.88 to 1.00 across items. 

The two researchers conducting the observations began each playground observation in zone 1 and then moved in order through the remaining eight zones. Observations were conducted at a designated location per zone to ensure both researchers were observing children using the playground from the same viewpoint. Each researcher scanned the target zone for a total of three to five minutes, depending on how busy the playground was, before moving on to the next target zone. The researcher conducted observations from left to right continuously for the duration of the three to five minutes. One entire scan of the playground took approximately 45 min to complete.

Institutional research ethics board approval was obtained for this study. The municipality in which the playground was located also granted the research team permission to undertake observations. During each observational session, the researchers had a one-page information letter describing the details of the research, information on the institutional research ethics board approval, and the contact information of the primary investigator. A verbal explanation of the study’s purpose and methods was also provided by the researcher onsite to any interested community members.

#### 2.2.3. Observation Schedule

All observations took place prior to the onset of the COVID-19 pandemic between October 2019 and November 2019 (average temperature of 12 degrees Celsius). An observation schedule was used based on Cohen et al.’s [44] guidelines for administering the SOPARC tool. According to these guidelines, a five-day schedule (used in the present study) should include observations at least twice a day: once in the morning (between 6:30 a.m. and 12:30 p.m.) and once in the afternoon (between 1:30 p.m. and 8:30 p.m.).

### 2.3. Statistical Analysis

Observations from each zone were grouped into their respective playground section (i.e., music/sensory, early childhood, freestanding, and main structure). Total counts of child observations from each of the four playground sections were then used for the final analysis. Descriptive statistics (frequency and proportions) were computed in R for gender (male and female), age group (ages 0–5 and ages 6–12, ages 13+), use (or not) of a wheelchair, mobility and/or visual aid, play behaviours (constructive, dramatic/fantasy, functional, games with rules, non-play, waiting) and social interaction types (peer play, play with an adult, solitary and parallel) observed across the overall playground as well as within each of the four playground sections.

## 3. Results

### 3.1. Observations

A total of 1332 child observations were made over the study period. Observations were conducted on eight different days for a total of 28 playground scans. Of the 28 total scans, 20 were completed in the afternoon and eight were conducted in the morning. Morning visits were conducted between 10:00 a.m. and 12:00 p.m. and afternoon visits were conducted between 12:30 p.m and 5:00 p.m. Nine scans were completed on the weekend and five scans were conducted on a Professional Activity (PA) day whereby children had the day off of school. The remaining 14 scans were conducted on a regular weekday.

### 3.2. Description of Playground Usage by Use of a Wheelchair, Mobility or Visual Aid, Gender and Age Group

Table 2 presents a complete description of playground usage by gender, age group, activity level, play behaviours and social interaction types. Of the 1332 child observations, one observation was made of a child who used a mobility aid (observed engaging in functional play with an adult within the main structure region). In terms of observed gender, there was a relatively equal number of female (51.82%) and male (48.20%) observations made on the playground. When broken down by playground section, there were more females observed in both the music/sensory and freestanding play elements sections than males.

When examining the age group of the children using the playground, most children observed were coded as being between the ages of 0–5 years (44.14%) and 6–12 years (51.80%). The greatest percentage of children aged 0–5 years was observed within the early childhood section. Most children aged 6–12 years were observed within the main structure section and children 13 years and older were mainly observed in the freestanding play equipment section.

### 3.3. Activity Level

Active play accounted for the majority of observations in all playground sections except for the freestanding play equipment section where the greatest percentage of play was non-active. The highest amount of very active play was observed on the main structure section. For a complete breakdown of activity levels across the playground and within each section, see Table 2.

**Table 2 ijerph-19-13648-t002:** Description of playground usage by observed gender, age group, interaction type and play behaviour type across the full playground and within each playground section.

		Playground Sections			
	Full Playground (*n* = 1332)	Music/Sensory (*n* = 116)	Early Childhood (*n* = 205)	Freestanding Play Equipment (*n* = 214)	Main Structure (*n* = 797)
**Gender ***					
*Male*	642 (48.2)	40 (34.5)	104 (50.7)	88 (41.1)	410 (51.4)
*Female*	690 (51.8)	76 (65.5)	101 (49.3)	126 (58.9)	387 (48.6)
**Age Group**					
*0–5*	588 (44.1)	65 (56.0)	119 (58.1)	93 (43.5)	311 (39.0)
*6–12*	690 (51.8)	48 (41.4)	78 (38.1)	97 (45.3)	467 (58.6)
*13+*	54 (4.1)	3 (2.6)	8 (3.9)	24 (11.2)	19 (2.4)
**Activity Level**					
*Non-Active*	273 (20.5)	41 (35.3)	63 (30.7)	106 (49.5)	63 (7.9)
*Active*	941 (70.7)	71 (61.2)	127 (62.0)	91 (42.5)	652 (81.8)
*Very Active*	118 (8.9)	4 (3.5)	15 (7.3)	17 (7.9)	82 (10.3)
**Play Behaviour Type**					
*Functional*	1175 (88.2)	102 (87.9)	172 (83.9)	190 (88.8)	711 (89.2)
*Dramatic/Fantasy*	35 (2.6)	2 (1.7)	16 (7.8)	0 (0)	17 (2.1)
*Games with Rules*	44 (3.3)	2 (1.7)	4 (2.0)	2 (0.9)	36 (4.5)
*Constructive*	3 (0.2)	3 (2.6)	0 (0)	0 (0)	0 (0)
*Non-Play*	71 (5.3)	7 (6.0)	13 (6.3)	21 (9.8)	30 (3.8)
*Waiting*	4 (0.3)	0 (0)	0 (0)	1 (0.5)	3 (0.4)
**Interaction Type**					
*Peer Play*	636 (47.7)	47 (40.5)	108 (52.7)	129 (60.3)	352 (44.2)
*Play with an Adult*	352 (26.4)	42 (36.2)	62 (30.2)	49 (22.9)	199 (25.0)
*Solitary*	292 (21.9)	27 (23.3)	25 (12.2)	26 (12.1)	214 (26.9)
*Parallel*	52 (3.9)	0 (0)	10 (4.9)	10 (4.7)	32 (4.0)

Note. Results presented as the proportion of observations on the full playground and within each section; count (%). * While we have coded gender using the terminology from the original SOPARC tool, we recognize that gender is a social construct that exists on spectrum and thus we use these terms with caution.

### 3.4. Playground Behaviour Types

Across the four playground sections, functional play occurred most often, accounting for 80–88% of the observed play behaviours (Table 2). Dramatic/fantasy play accounted for less than 10% of play behaviours observed in each of the four sections, with the highest percentage of dramatic/fantasy play observed in the early childhood section (7.8% of play behaviour in this section). Constructive play was only observed in the music/sensory section (2.6% of observed play behaviours in this section). Waiting was only observed in the freestanding play equipment and main structure sections and accounted for less than 1% of behaviours observed in these two sections.

### 3.5. Social Interaction Types

The most common social interaction type across all four playground sections involved peer (child/child) play (Table 2). Most of the peer play was observed within the freestanding play equipment section (60.3% of all observations). At 36.2% of all observations, the music/sensory section elicited the most interactive play between children and adults (e.g., parent, caregiver). The main structure section was observed to produce the most solitary play (26.9%) while most instances of parallel play were observed primarily in the early childhood section (4.9%).

## 4. Discussion

To our knowledge, this is the first study to involve naturalistic observations of children’s usage and types of play on a playground designed to be accessible and inclusive. While only one child was observed to use a wheelchair, mobility aid or visual aid, the results of this study provide important information on how children use a playground designed to be accessible and inclusive for children of all genders, ages and abilities. Our results demonstrate that this inclusive playground is highly utilized by both males and females of various ages. Results show that this playground supports interactive and individual play, as well as some play behaviours beyond functional play. These results also demonstrate that most children on the playground engage in either active or very active play.

### 4.1. Demographics and Activity Levels of Children Using the Playground

Outdoor play and physical activity have both been shown to enhance several aspects of child health including physical fitness, motor development, cognition and mental health outcomes [11,13,14,23,45]. Currently, only 28% of Canadian children and youth are reaching the recommended levels of physical activity [46], and girls are consistently shown to engage in less physical activity compared to boys [47,48,49]. Thus, it is important to continue to determine strategies to encourage and enhance play, and to make play, particularly outdoor play, accessible for all children [47,50]. Our results demonstrate that a playground designed to be accessible and inclusive for all ages and abilities affords children the opportunity to play and be active outdoors. Across the eight study observation periods, 1332 instances of play were captured. Of the total child observations made, an approximately equal number of observations were made of males and females across the full playground, and 80% of children observed were engaging in active or very active play. These findings align with previous studies showing that in recent years, boys and girls are participating and being active on playgrounds at a near-equal level [51,52]. Interestingly, the music/sensory section of the playground elicited the greatest percentage of observations made of females, accounting for nearly 70% of all child observations made in this section. To date, most playground research has identified features of playgrounds that deter girls from participating (e.g., ball games, large paved areas) [51,52]. The findings from this study build on previous literature by identifying aspects of playgrounds that may be of particular interest to females (i.e., music and sensory elements). Thus, not only do these results provide evidence that an inclusive playground is generally supporting active play for males *and* females equally, but music and sensory elements may be particularly good at enhancing outdoor play among females.

In addition to supporting outdoor play among males and females, our findings also suggest that the playground was attracting children across different age groups. Most children appeared to be under the age of 12, however, there were several instances where children may have been between 13 to 17 years old. The freestanding play equipment section was highly used by older children, with 12% of the observations in this section being children aged 13 and older. Within the literature, a design recommendation for inclusive playgrounds is to ensure an appropriate level of challenge to engage children of all abilities [32]. Our study observations included children in the preschool years, school-aged children, and even children in early adolescence. Thus, observing children from all age groups may suggest that this inclusive playground facilitates play among various age groups. That is, thinking developmentally, children of different ages will possess different abilities, and this playground space appears to offer opportunities for play that were of interest to all children we observed.

The playground observed in this study was physically accessible and aligned with many of the design recommendations for inclusive playgrounds [32,53] and universal design [33]. In terms of its physical accessibility, the playground included a rubberized surface, double-wide ramps that led onto the main playground section from multiple access points and various adapted pieces of equipment including adapted swings and slides with transfer benches. Further, the playground included elements of sensory play within a section devoted to sensory and music elements and included features such as the ‘cozy dome’ to provide a retreat from overstimulation. Scholars have previously noted that inclusive playgrounds are often segregated spaces and may not necessarily produce opportunities for integrated play between children with and without disabilities [54]. Consistent with the goals of universal design [55], an important finding from the current study is that this inclusive playground appeared to attract a large number of children who did not use a wheelchair, mobility or visual aid, and/or children who may live with impairment or disability in a way that is not immediately observable or visible. While only one child was observed to use a mobility aid, findings show promise that a playground built in accordance with the universal design principles is supporting various play opportunities among males and females of all ages. Accessibility and inclusion on playgrounds to date has largely focused on the physical accessibility of playgrounds [33]. From a universal design perspective, playgrounds must be designed to go beyond physical accessibility and offer children of different ages, genders and abilities opportunities to engage in various types of play behaviours and social interactions [33]. Designing playground spaces that meet the universal design principles is imperative to ensuring inclusion and protecting all children’s right to play [2]. However, universal design may not be a sufficient approach for encouraging playground play among children who use wheelchairs, mobility devices, and/or visual aids. More empirical work is needed to better understand how children with disabilities engage with playgrounds designed to be inclusive and accessible.

### 4.2. Play Behaviours

Similar to previous research on traditional and contemporary playgrounds [56,57], we found that functional play accounted for a greater percentage of all play, across all playground sections, when compared with dramatic/fantasy and constructive play, as well as waiting and non-play. Functional play includes activities like running, wheeling, jumping, swinging and climbing, all of which promote children’s physical development [58]. Our findings suggest that the playground observed in this study is particularly good at promoting functional play in children, which aligns with the higher activity levels that were observed among children on the playground. These findings may relate to the size of the playground observed in this study and the variety of equipment available. Additionally, this playground consisted of open spaces for children to walk, run and wheel around the playground all of which are associated with greater activity levels in children.

While the playground was observed to support functional play, few obvious markers of dramatic/fantasy or constructive play were observed. This finding aligns with previous research on playgrounds that reported little dramatic/fantasy and constructive play in relation to total playground behaviours [41,56,57]. The incorporation of play components that offer manipulation and differential feedback (e.g., sand and water play components, loose play pieces) could help to produce more constructive play opportunities [59]. Further, incorporating loose parts pieces (such as sand toys, and sport balls), musical instrument equipment, or playground features shaped in recognizable designs (e.g., a house or boat) could support more imaginative play [32,41,60,61,62]. The playground did include a variety of musical instrument equipment, yet there were still very few instances of dramatic/fantasy play and constructive play observed. This finding is contrary to previous research suggesting that music equipment may increase dramatic and constructive play in children [41]. It is possible that including fixed musical equipment may not be enough to encourage other types of play behaviours beyond functional play without the incorporation of other loose parts play equipment. The music/sensory section was also located on the periphery of the playground which could be beneficial to reduce noise and sensory stimulation. However, the location may have resulted in this section being less ‘seen’ by children as they were located on the perimeter of the playground which could have impacted children’s awareness of what this section had to offer. Future research may consider the impact of raising users’ awareness, such as through informational materials (e.g., brochures, way-finding signage) of what is available within inclusive playground spaces.

Loose parts play pieces or sand/water play components were not available at this playground; rather, it was comprised of fixed play structures that primarily supported gross motor skill activities (e.g., climbing, swinging, sliding). The lack of loose parts on this playground is likely attributable to the fact that implementing play equipment with loose parts can be difficult. Key challenges include the risk of losing equipment and having adequate equipment storage space [62]. To support loose parts play on or nearby inclusive playgrounds in the future, it may be useful to consider providing locked storage spaces (i.e., for loose parts play components) that could be accessed by playground staff and/or local community organizations that use the playground for play programming. Ensuring that any loose parts play options are accessible also requires attention. For example, raised sand tables are becoming increasingly more common structures within playgrounds because they allow children using wheelchairs (or other mobility devices) to access sand play [63]. When designing and locating such raised sand tables, attention should be given to preventing the sand from spilling onto and causing damage to playground surfaces (e.g., by spacing a raised sand table from the main playground surface and/or using an alternative playground surface around the raised sand table).

### 4.3. Social Interactions

Play with peers and adults are associated with positive developmental trajectories [8,10]. Our finding that the majority of children were observed interacting with either peers or an adult highlights the opportunity the inclusive playground may provide for supporting child development through interactive play [7,8]. Peer play in particular has been shown to provide additional benefits for social development and school readiness [9,10]. We observed greater peer play occurring within the freestanding play equipment section, with peer play accounting for nearly 60% of all observations within this section. This finding may be attributable to the play equipment within the freestanding section being intentionally designed to promote play among pairs of children or small groups. For example, the swings structure included a ‘friendship’ swing that allows for two children to swing together while facing one another, a rotating climbing globe structure that can be used by many children at the same time and see-saws that support four users rather than two. Our findings suggest that including play equipment pieces designed for partner play or small groups within playgrounds could enhance opportunities for development through peer play.

In addition to peer play, it is important that playgrounds provide opportunities for children to engage in solitary and/or parallel play [32]. Playgrounds including elements that support solitary play have been shown to help children avoid and cope with overstimulation [63,64]. These opportunities of solitary play are of particular importance to children who may prefer playing in spaces that are quieter and more private [63,64]. Within the current study, instances of solitary play accounted for 22% of observations made with parallel play accounting for 4%. These findings provide some evidence that this playground is supporting inclusion through providing opportunities for both solitary and parallel play.

### 4.4. Limitations and Future Directions

While this study’s naturalistic observational approach offers valuable information, it has some limitations that should be considered when interpreting the results. First, although the inter-rater reliability for all variables coded were considered good or excellent as per Koo and Li’s guidelines [43], some information coded may not have reflected the true nature of the child’s intended play behaviour. For example, it is possible that children engaging in dramatic/fantasy play while running around the playground may have been coded as ‘functional’ play if the researcher did not hear the imaginative dialogue occurring between the children. Future research should obtain more nuanced and contextual information regarding play instances by performing in-depth analyses of play (e.g., via analysis of video recordings of playground play, and/or analysis of qualitative data focused on how children use playgrounds). Our findings are also based on singular observations made by one researcher at a time. As such, it is possible that instances of play may have been missed or went unaccounted for as a child transitioned between play types. To address this limitation, future research could consider having multiple observers present and/or video-record playground play.

Our research design did not allow for certain demographic information to be collected (e.g., impairment type). While a new code was added to the SOPARC tool to allow for observations of children’s use of a wheelchair, mobility and/or visual aid, we acknowledge that this coding does not capture the full extent of childhood disability. Specifically, the observational nature of our study did not allow for children who live with other types of disabilities that are not observable to be coded. Future research should consider ways to observe play among children with a wide range of disabilities including those that are visible and invisible.

Further, while we only observed one child who used a mobility device (i.e., lower leg prosthetics), there were likely some children with other types of disabilities (e.g., invisible disabilities) who went unaccounted for given the limitations of the coding tool. Within Canada, 3.6% of children ages 0–14 years-old experience disability [63], suggesting that there would be fewer children on any one playground observed to have a disability [65]. This statistic, combined with our coding of disability within the SOPARC tool may explain why few children were observed to use a wheelchair, mobility device or visual aid. Future research is needed to confirm these results as well as to develop appropriate coding methods to account for children’s range of abilities.

The playground was also fairly new, opening just one year prior to our study. The ‘grand opening’ event occurred only a few months before the observations occurred. As a result of its relatively recent installation, it is possible that families of children with disabilities were unaware of the new playground or were not familiar with its intentional inclusive design. Observing few children who used a wheelchair, mobility and/or visual aid on this fairly new inclusive playground highlights the importance of municipalities communicating the development of accessible and inclusive playgrounds and perhaps creating and/or promoting programming options to help ensure that the community is aware of its presence and the intent of its design. Municipalities might consider having playground staff on-site at inclusive playgrounds during scheduled hours to support different play programs, aid caregivers with lifts and transfers, and/or to offer parents information about the playground’s various inclusive design elements [31,66]. This could make the playgrounds more inclusive for children with disabilities and their families and, in turn, may increase their usage of inclusive playgrounds.

## 5. Conclusions

In order for all children, regardless of gender, age and ability, to access and engage in play on community playgrounds, they must be designed with inclusion in mind. This means that in addition to physical accessibility, playgrounds should include play equipment that is intentionally selected and positioned to provide a variety of play opportunities for children with different play preferences and abilities. Our findings demonstrate that the observed accessible and inclusive playground was supporting play among males and females, of which the majority were 12 years old or under. The study findings also demonstrate that the observed playground supported opportunities for a variety of social interactions and, to some extent, play behaviours beyond functional play. More research is needed to better understand the lack of children with a visible disability observed within the playground. These findings highlight that playgrounds that are intentionally designed for accessibility and inclusion are being utilized within communities. Future research is warranted on how play leaders, recreational programmers, and educators can utilize the design features of accessible and inclusive playgrounds to support equitable play experiences for children of all ages, genders and abilities.

## Figures and Tables

**Figure 1 ijerph-19-13648-f001:**
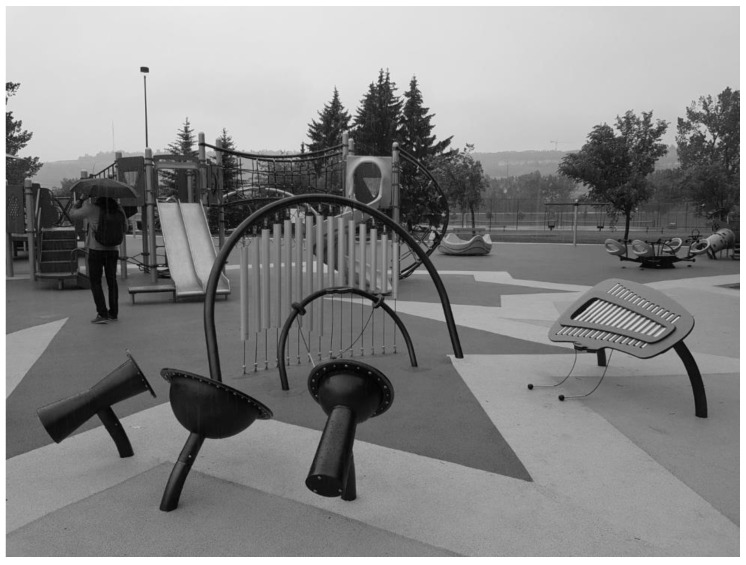
Image depicting part of the music/sensory section of the observed playground.

**Figure 2 ijerph-19-13648-f002:**
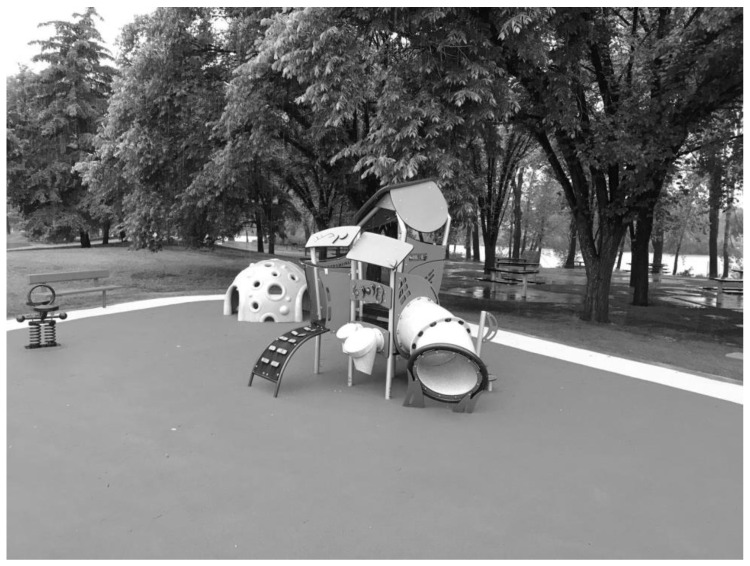
Image depicting part of the early childhood section of the observed playground.

**Figure 3 ijerph-19-13648-f003:**
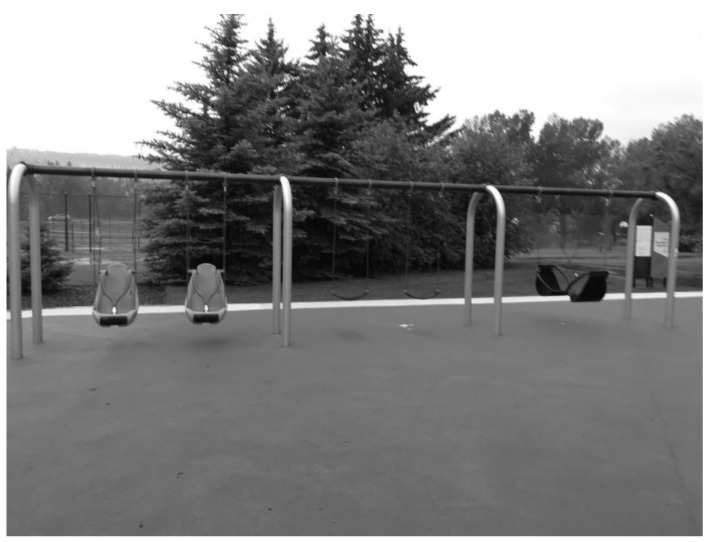
Image depicting part of the freestanding equipment section of the observed playground.

**Figure 4 ijerph-19-13648-f004:**
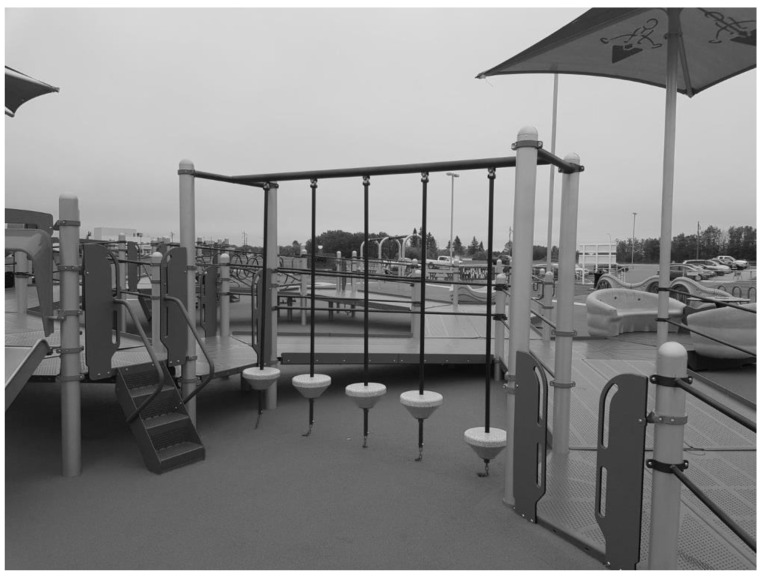
Image depicting part of the main structure section of the observed playground.

**Table 1 ijerph-19-13648-t001:** Definitions of Codes Added to the Modified SOPARC Tool.

Code	Definition
**Assistive Device/Aid**	
Wheelchair	Any seated device utilized by a person with a mobility disability for the purpose of locomotion [42].
Mobility Aid	Any non-seated locomotion aid including, but not limited to, braces, crutches, canes, and walkers [42].
Visual Aid	Use of a device, or animal for the purpose of visual guidance (e.g., white cane, guide-dog, peer).
None	No visible use of a wheelchair, mobility or visual aid.
**Activity Level ^a^**	
Non-Active	Activities that require little physical effort (e.g., lying down, sitting, standing).
Active	Activities that require a moderate amount of physical effort (e.g., walking, wheeling).
Very Active	Activities that require a great deal of physical effort (e.g., running, jumping, racing).
**Play Behaviour ^b^**	
Constructive	Child’s activity is goal-oriented and thoughtful. They are using materials to create something (e.g., using rocks to make a structure).
Dramatic/Fantasy	Child takes on imaginary roles or uses objects to represent something imaginary (e.g., children playing “house” or pretending to be animals).
Functional	Play activities involving repetitive muscle movements (e.g., running, walking), vestibular stimulation (e.g., rocking back-and-forth, swinging, jumping, spinning, rolling on the ground), or proprioceptive stimulation (e.g., climbing, pushing, pulling, carrying heavy objects).
Games with Rules	Games with universal rules such as tag, dodgeball, hide- and-go-seek.
Non-play	Child is not involved in any of the above play behaviours. Examples of non-play behaviours include unoccupied/onlooker play (i.e., watching others), being between activities, and sitting.
Waiting	Child is not engaged in play because they are waiting to use equipment (e.g., waiting to use the swing).
**Social Interaction ^b^**	
Peer Play	Instances when two or more children are playing in an activity-oriented way and mutually acknowledging the other(s). The children’s actions are complementary with those of another/others, and/or the children are engaged in conversation about a common activity.
Play with an Adult	Instances where a child is engaging in play with an adult (e.g., parent, caretaker).
Solitary	Instances where a child plays alone or independently, makes no reference to others and makes no effort to include other children in his or her play.
Parallel	Instances where a child plays independently beside, but not with, another child. Child does not try to influence others in play.

^a^ Definitions of activity level based on McKenzie and Cohen [38], ^b^ Definitions of play behaviours and social interactions based on Maxwell et al. [41].

## Data Availability

For more information regarding data availability, please email K.P.A.-N., kelly.arbour@utoronto.ca.

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
