# Peer review of "Children’s Usage of Inclusive Playgrounds: A Naturalistic Observation Study of Play"

_ijerph, 2022, doi:10.3390/ijerph192013648_

Round 1

Reviewer 1 Report

It is, in general, a very well planned and resolved article.

The central topic of the research article: Children’s Usage of Inclusive Playgrounds: A Naturalistic Ob- 2 servation Study of Play is of great interest although it has been approached from a very particular context and the results are not discussed with previous studies in this regard.

The introduction has few significant references. In some places it seems more like a manifesto or an opinion piece than an element of support for the arguments presented below. Once again, the topic is quite relevant, but it lacks more support and a better connection between the different sections of the document. In the introduction authors can explain their motivations and why the issue is relevant.

The list of references is short. I suggest the authors to expand the literature review and provide more concrete results from previous research.

The literature review must be orderly: ideas are not connected. I suggest to group ideas and create a logic discourse.

Data collection and analysis must provide more insight in how the data was analysed.

The paper must finish with a proper conclusions and not with a summary.

Author Response

Thank you for taking the time to provide valuable feedback on our paper. We believe that by addressing yours, and the other reviewers, feedback the clarity and organization of our paper has been strengthened. Please see below for a point-by-point response to each of your comments.

COMMENT: The central topic of the research article: Children’s Usage of Inclusive Playgrounds: A Naturalistic Observation Study of Play is of great interest although it has been approached from a very particular context and the results are not discussed with previous studies in this regard.

RESPONSE: We have now added more literature regarding traditional playgrounds in the introduction (see response below for specific examples). We also added one paper to the introduction that discusses the usage of inclusive playgrounds in Switzerland (Wegner et al.,2021).

COMMENT: The introduction has few significant references. In some places it seems more like a manifesto or an opinion piece than an element of support for the arguments presented below. Once again, the topic is quite relevant, but it lacks more support and a better connection between the different sections of the document. In the introduction authors can explain their motivations and why the issue is relevant.

RESPONSE: We have re-structured the introduction to improve clarity regarding the purpose of the study. In particular, we have highlighted that the selection and layout of play equipment have been shown to influence play behaviours in children on traditional playgrounds. This evidence suggests that certain play equipment may facilitate/hinder participation among certain groups of children (e.g., boys vs. girls, children with vs. without a disability) and provides the bases for which inclusive playground design was built upon. With an increase of playgrounds intentionally design to be inclusive, it is necessary to examine how these playgrounds are being used and by whom. This was the impetus for the current study. We hope that the way we have re-structured the introduction helps to address the noted concern here.  

COMMENT: The list of references is short. I suggest the authors to expand the literature review and provide more concrete results from previous research.

REPSONSE: Building upon our response above, we have added some more literature on the observational research that has been conducted to date on traditional playgrounds. For example, we added literature conducted on traditional playgrounds describing the differences in play behaviours among boys and girls (Hidnman et al., 2015; page 2 lines 67-70). We also discussed a recent study (Wegner et al., 2021) that explored children’s experiences on inclusive playgrounds in Switzerland. This addition can be found on page 3 lines 194-197. The addition of this literature provides the reader with a richer understanding of how traditional playgrounds have been shown to be used. The discussion about the Wegner et al., 2021 paper provides readers with an overview of the only study, to our knowledge, that examined the usage of an inclusive playgrounds and helps to strengthen the argument that more research in the field of inclusive playgrounds is necessary, especially in country-specific contexts.

COMMENT: The literature review must be orderly: ideas are not connected. I suggest to group ideas and create a logic discourse.

RESPONSE: Thank you for bringing this to our attention. We have spent some time re-organizing the introduction (see responses above) to improve flow and to create a logic discourse. Now, we provide an overview of the importance of play. research on traditional playgrounds and conclude with a discussion of what inclusive and accessible playgrounds are and why they are important. 

COMMENT: Data collection and analysis must provide more insight in how the data was analysed.

RESPONSE: Given that the purpose of this study was to describe how the playground was being used and by whom, the analysis consisted of descriptive statistics of counts within different playground sections (e.g., early childhood, main structure etc.). The analysis is described on page 8. To improve clarity, we added a description of the codes we were conducting descriptive statistics for on page 9 lines 358-360 (e.g., for social interactions we included the codes peer play, play with an adult, ssolitary,and parallel play). We are not clear on what other level of detail the reviewer is requesting be added to this section beyond the detail that currently exists within the statistical analysis section. We would be happy to consider adding more detail in this section to enhance its clarity if further information can be provided.

COMMENT: The paper must finish with a proper conclusions and not with a summary.

RESPONSE: We have re-written the conclusion based on your comment here. We hope that this addresses your concern.

Reviewer 2 Report

While this paper is well-written and organized, I think the topic itself does not contribute to the literature much. Overall, how children use a playground is covered and known in the literature. The necessity and contribution of this paper are questionable.

Author Response

Thank you for taking the time to read and provide feedback on our paper. We believe that by addressing yours, and the other reviewers, feedback the clarity and organization of our paper has been strengthened. Please see below for our response to your comment.

Reviewer 2

COMMENT: While this paper is well-written and organized, I think the topic itself does not contribute to the literature much. Overall, how children use a playground is covered and known in the literature. The necessity and contribution of this paper are questionable.

RESPONSE:

We can appreciate that there has been research conducted on children’s play behaviour on traditional playgrounds. However, research on playgrounds to date have largely ignored children with disabilities (e.g., Hyndman et al., 2015; Broekhuizen et al., 2015; Dowda et al., 2009). The reason for this may be two-fold. First, observational tools that exist to observe physical activity and play do not include variables to observe if a child has a visible disability. To address this, we modified the SOPARC tool to include an item on disability. Second, traditional playgrounds (for which most of the research conducted to date has examined) are not designed to include children of all abilities. Our study focused specifically on a playground intentionally designed to be accessible and inclusive for children of all ages and abilities (which in Canada, where this study occurred, are very limited in their availability) to examine, through naturalistic observation, what groups of children come to the playground and how the playground is being used. In order to include children with disabilities without intervening (i.e., asking for demographic information) or by bringing children with disabilities specifically to that playground (which goes against naturalistic design, our observations were limited to visible disabilities (i.e., of children who use a wheelchair, mobility device and/or a sensory aid such as a guide dog). We only observed one child who used a mobility aid (i.e., leg braces). Therefore, while the purpose of this study was to examine play behaviours of children with and without disabilities, we were limited based on the single observation of a child with a disability (as defined within the coding we used). This is a finding in and of itself and we discuss the reasons why only one child with a visible disability may have been observed and future directions to increase participation of children with disabilities on playgrounds that are intentionally designed to be accessible and inclusive. This discussion is included on page 12 lines 502-508 and page 13, lines 717-752. Despite this, we believe our paper has important contributions to the literature regarding play behaviours on inclusive playgrounds. Inclusive playgrounds are designed to include children of all ages, genders and ability levels. Our findings highlighted that perhaps children with visible disabilities are not using the playground as much, however, we did find that the playground supported play among boys and girls and across different age groups. Our paper also provides important methodological considerations for future observational research on inclusive playgrounds regarding how to modify observational tools in order to gain some information on disability.

To enhance the paper’s writing clarity, we have re-organized the introduction and re-written the purpose statement. We hope that in doing so, the importance and necessity of this work is clear to the reader. Our purpose statement now reads: “The purpose of this study was to examine how a newly built playground intentionally designed to be accessible and inclusive of children of all ages, gender and abilities is being used by the community and by whom.”

Reviewer 3 Report

This is a good, and very publishable article overall but the emphases need to be 'tweaked.' From the title and introduction, I thought I was going to read a study that focused on children with disabilities using a playground designed to include them, but it turns out that only one child with a disability was observed. The other data is interesting in itself, but it needs to be framed and introduced in a way in which the reader realises that this is a report on non-disabled children using the playground. It may be that you could later set up an observation of children with a range of disabilities accessing the playground alongside other children?

Author Response

Thank you for taking the time to provide valuable feedback on our paper. We believe that by addressing yours, and the other reviewers, feedback the clarity and organization of our paper has been strengthened. Please see below for a point-by-point response to each of your comments.

Reviewer 3

COMMENT: This is a good, and very publishable article overall but the emphases need to be 'tweaked.' From the title and introduction, I thought I was going to read a study that focused on children with disabilities using a playground designed to include them, but it turns out that only one child with a disability was observed. The other data is interesting in itself, but it needs to be framed and introduced in a way in which the reader realises that this is a report on non-disabled children using the playground. It may be that you could later set up an observation of children with a range of disabilities accessing the playground alongside other children?

RESPONSE:

We appreciate your interest and recognition of the importance of this research. The intent with this work was to explore who uses playgrounds designed to be inclusive and accessible for the community (e.g., gender, age, disability group) and to understand how children of different genders, age groups and abilities were using this playground. To make this study purpose clearer, we have re-organized the introduction to try and shift the focus from being solely on children with disabilities to a focus on inclusion more broadly (e.g., inclusive to boys and girls, inclusive to children of various age groups and children with different play preferences). We also re-worded our purpose statement to improve clarity on this.

This playground was designed to support play among all genders, age groups and had a particular emphasis on play among children with disabilities. As such, we too expected that we would be able to report on the play behaviours among children with disabilities on this playground. In order to include children with disabilities without intervening (i.e., asking for demographic information) or by bringing children with disabilities specifically to that playground, we were limited to disabilities that could be observed (i.e., using a wheelchair, mobility device or sensory aid). Unfortunately, we only saw one child who was coded as using a mobility device (i.e., leg braces). While this small sample size limits our ability to describe the play behaviours of children with disabilities – which was what we set out to do a priori – this is an important finding in and of itself. We discuss the reasons why only one child with a visible disability may have been observed and future directions to increase participation of children with disabilities on inclusive playgrounds. This discussion is included on page 12 lines 502-508 and page 13, lines 717-752. Despite this, we believe our paper has important contributions to the literature regarding play behaviours on inclusive playgrounds. Inclusive playgrounds are designed to include children of all ages, genders and ability levels. Our findings highlighted that perhaps children with visible disabilities are not using the playground as much, however, we did find that the playground supported play among boys and girls and across different age groups. Our paper also provides important methodological considerations for future observational research on inclusive playgrounds regarding how to modify observational tools in order to gain some information on disability.

Round 2

Reviewer 2 Report

I appreciate the authors' efforts to revise the paper. The authors now clearly explained the purpose of the research and provide suggestions for future studies.